# Recent Advancements in Polysulfone Based Membranes for Fuel Cell (PEMFCs, DMFCs and AMFCs) Applications: A Critical Review

**DOI:** 10.3390/polym14020300

**Published:** 2022-01-12

**Authors:** Rajangam Vinodh, Raji Atchudan, Hee-Je Kim, Moonsuk Yi

**Affiliations:** 1Department of Electronics Engineering, Pusan National University, Busan 46241, Korea; vinoth6482@gmail.com; 2Department of Chemical Engineering, Yeungnam University, Gyeongsan 38541, Korea; atchudanr@yu.ac.kr; 3Department of Electrical and Computer Engineering, Pusan National University, Busan 46241, Korea

**Keywords:** polysulfone, polymer electrolyte membrane, DMFCs, AMFCs, sulfonation, Nafion^TM^, fillers, inorganic/organic hybrid membranes

## Abstract

In recent years, ion electrolyte membranes (IEMs) preparation and properties have attracted fabulous attention in fuel cell usages owing to its high ionic conductivity and chemical resistance. Currently, perfluorinatedsulfonicacid (PFSA) membrane has been widely employed in the membrane industry in polymer electrolyte membrane fuel cells (PEMFCs); however, Nafion^TM^ suffers reduced proton conductivity at a higher temperature, requiring noble metal catalyst (Pt, Ru, and Pt-Ru), and catalyst poisoning by CO. Non-fluorinated polymers are a promising substitute. Polysulfone (PSU) is an aromatic polymer with excellent characteristics that have attracted membrane scientists in recent years. The present review provides an up-to-date development of PSU based electrolyte membranes and its composites for PEMFCs, alkaline membrane fuel cells (AMFCs), and direct methanol fuel cells (DMFCs) application. Various fillers encapsulated in the PEM/AEM moiety are appraised according to their preliminary characteristics and their plausible outcome on PEMFC/DMFC/AMFC. The key issues associated with enhancing the ionic conductivity and chemical stability have been elucidated as well. Furthermore, this review addresses the current tasks, and forthcoming directions are briefly summarized of PEM/AEMs for PEMFCs, DMFCs, AMFCs.

## 1. Introduction

### 1.1. Fuel Cells

The fuel cell is an electro-chemical energy conversion design; it alters the chemical energy of the reactants directly into electric energy along with heat and potable water. As global environmental and energy issues become more and more acute, incredible efforts are being made to explore new energy selections. As a new energy technology, fuel cells have been shown to be highly efficient and have an excellent ability to convert conventional fossil fuel energies due to low or zero-emission [1,2,3]. Fuel cells and batteries share multiple similarities: both are based on the anode-to-cathode electronic transfer principle and convert chemical energy into electric energy; they both require an electrolyte and external load to perform useful work and generate low DC voltages. Fuel cells are stacked similarly to batteries as well. Extensive power and voltage output is achieved by combining many cells in series. The main differences between fuel cells and batteries are the nature of their electrodes. Batteries use metallic anodes (lithium or zinc) and cathodes (generally metallic oxides). During operation, batteries consume the anode and the cathode, which will need recharge or replacement. In contrast, fuel cells operate with externally supplied reactants and do not consume any part working part of the cell. Therefore, fuel cells need no recharge and can continue operating as long as the reactant is supplied. Such repeated charging and discharging resulted in decreasing the life-time of the battery compared to that in the case of the fuel cell. In addition, fuel cells provide an inherently clean source of energy, with no adverse environmental impact during operation, as the byproducts are simply heat and water [4]. Nevertheless, the recent constraint in fuel cell commercialization stalks from the expensive nature of the raw materials (Nafion^TM^ electrolyte membrane and noble metal catalysts) and of the manufacturing method [5]. Furthermore, fuel cell electric vehicles (FCEV) are under progress by many automobile companies and are effectively verified due to their several advantages. Even though many advantages are in the fuel cells, there is a gap in the implementation of the fuel cells in on-road vehicles due to some practical issues.

### 1.2. Types of FCs

Fuel cells are divided into direct and indirect fuel cells according to their working temperature, the fuel cell components, and the type of electrolytes used. Fuel cells are classified according to their operational temperatures, such as low-temperature fuel cells and high-temperature fuel cells, as shown in Figure 1a [6]. The PEMFC, AMFC, and DMFC belong to low-temperature fuel cells. Molten carbonate fuel cell (MCFC), phosphoric acid fuel cell (PAFC), and solid oxide fuel cell (SOFC), fall into the high-temperature fuel cells group. In this cataloging, PEMFCs are more promising and consistent than other fuel cells owing to their versatile applications, extraordinary efficacy, and tiny emission of impurities, and can be the basis for DMFCs and AFCs. The acidic or alkaline concentrations are applied as electrolytes in fuel cells termed as mobile electrolyte systems, whereas electrolytes are immersed in a porous-based (pores enriched) material, defined as an inert/immobile electrolyte system or matrix system [7,8,9,10].

#### 1.2.1. PEMFCs

In a typical PEMFC, the cation exchange membrane (CEM or PEM) is accountable for the proton conductivity, which permits the passage of H^+^ from anode to the cathode, establishing the essential component of the electrochemical device. In various types of fuel cells, membranes constructed with perfluorinatedsulfonicacid (PFSA) is predominantly employed because of its excellent proton conductivity and adequate chemical/mechanical characteristics; they are worked at temperatures between 120 and 180 °C in high pressure [11,12]. The structure of PFSA is illustrated in Figure 1b; the sulfonic acid group is connected to the perfluoroethereal side chains of the PFSA. Proton conductivity is due to the significant phase separation between hydrophilic and hydrophobic domains in PFSA when hydrated; and the chemical/mechanical stability is caused by the rigid structure of the polytetrafluoroethylene (PTFE) backbone and strong C-F bond even in the side chains. However, this type of membrane exhibited a severe defect at temperatures below zero degrees Celsius and above hundred degrees Celsius [13,14]. Another kind of membrane, Nafion^TM^, developed and introduced by Dupont in the 1960s, has been extensively studied and is a commercially available proton-conducting membrane in PEMFC applications. Nafion^TM^ shows excellent characteristics, such as high electrochemical and chemical stability, low permeability to reactant species, selective and high ionic conductivity, and the ability to provide electronic insulation. However, the Nafion^TM^ membrane showed poor proton conductivity at higher temperatures due to dehydration of water, which controlled the number of water-filled channels [15,16,17]. To solve these problems, researchers have developed alternative ways of proposing other polymeric materials, such as SPEEK (sulfonated poly(ether ether ketone)) [18,19], polybenzimidazole (SPBI) [20,21], and polysulfone (SPSU or SPSF) [22,23]. These membranes show their strengths in different features of water uptake %, ionic conductivity, and mechanical and thermal stability. The pictorial illustration of PEMFC is depicted in Figure 2a along with cell reaction.

#### 1.2.2. DMFCs

The DMFCs have few merits of efficiently working at low temperature, easy strategy, and eco-friendly characteristics. The usage/handle of methanol is also easy since it exhibits liquid properties at ambient temperature. More specifically, unlike PEMFCs, aqueous methanol-based DMFCs do not require a humidification system and peculiar thermal management aids. They also have superior energy and power density as compared to indirect fuel cells and recently established lithium-ion batteries (LIBs). A plausible usage of the DMFC comprises portable electronic gadgets, military communications, transportation services, and traffic lights/signals [23,24,25]. The schematic of DMFC is presented in Figure 2c. The major issue with DMFCs is methanol cross-over as it permeates methanol along with water from anode to cathode direction. During DMFC operation, methanol cross-over outcomes in low power-output due to methanol oxidation at the cathode with the aid of cathode catalysts, leading to (i) electrode depolarization, (ii) mixed potential, consequently open-circuit voltage (OCV) of the DMFC less than 0.8 V, (iii) consuming of oxygen, (vi) CO poisoning, and (v) severe water accretion at the cathode, which restricts oxygen contact to cathode catalyst spots. In addition, the presence of excessive methanol cross-over lowers the overall performance of the fuel cell [26,27,28].

#### 1.2.3. AMFCs

In principle, AMFC is a feasible substitute for PEMFC and is currently receiving new consideration. In AMFCs, the AEM conducts OH^−^ (hydroxide) or CO_3_^2−^ (carbonate) anions while an electric current is flowing, which has numerous advantages, (i) in a high alkaline environment, both oxygen reduction reaction and methanol oxidation are more predominant; electro-osmotic drag by OH^−^ moves from cathode to anode, which reduces anode to cathode methanol cross-over, simplifies water management, and (ii) allows the use of non-noble metal catalysts. These AEMs are cheap and have improved mechanical/chemical characteristics compared to PEMs. In recent years, more research attempts have been performed to synthesize novel AEMs to enhance their ionic (OH^−^) conductivity [29] along with alkaline stability [30,31]. Polymer back bones such as polystyrene (ethylene butylene) polystyrene [32], poly(2,6-dimethyl-1,4-phenylene oxide) [33], polystyrene [34], poly(ether ether ketone) [35], poly(vinyl alcohol) [36], and polyether sulfone [37,38] have been expansively explored to synthesize alkaline membranes. The afore-mentioned polymeric materials can be readily functionalized with the following cationic groups, quaternary phosphonium [39,40], guanidinium [41,42], quaternary ammonium [43], or imidazolium [44,45,46] which are accountable for creating the polymer backbone conductive.

### 1.3. Ion Exchange Membranes

For all FCs, the membrane (PEM or AEM) is the heart of the FC. It plays a prominent part in the transportation of ions within a fuel cell via the following aspects: (1) friction through the pore walls, (2) the energy of the membrane swelling process, (3) complete blockage of transport due to insufficient water absorption, (4) hydrophobic/hydrophilic contact between solvation shells and water dipoles, (5) effects of double-layer and (6) surface diffusion [47,48]. The main difference between CEMs and AEMs are tabulated in Table 1.

### 1.4. Preliminary Characteristic of IEM

The prepared IEMs were subjected to the following preliminary characterization studies: water uptake (WU), ion exchange capacity (IEC), ionic conductivity, permeability of methanol (p), and alkaline stability test to check the appropriateness of the IEMs in FC applications, and the pictorial protocol is illustrated in Figure 3.

#### 1.4.1. Water Uptake and Schroeder’s Paradox

The WU of the IEM was measured by calculating the weights of the dry and wet membrane samples. The dry membrane weight (W_dry_) is obtained by drying the sample at 100 °C for 12 h immediately before weighing it. The weight of the corresponding membrane in wet conditions (W_wet_) is obtained by immersing the membrane sample in deionized water (DI water) at room temperature for about 1 day, wiping off the surface moisture with filter paper and then quickly weighing it. The water uptake (%) was determined from the subsequent equation [49]:(1)$$\mathrm{WU}\;\left(\%\right)=\frac{W_{\mathrm{wet}\;}-W_{\mathrm{dry}}}{W_{\mathrm{dry}}}\times 100\%$$

In addition, the sorption may be measured by bringing a membrane to equilibrium with a liquid by either immersion of the membrane into the liquid (directly) or by contact with the vapor phase (isopiestically). Since the solution, the vapor, and the sample are all in equilibrium, it is believed that there is no difference between the two methods. The uptake of water by PFSA from a liquid reservoir and a saturated vapor reservoir differs under the same conditions. This phenomenon is called Schroeder’s paradox, and more recently, attempts have been made to explain this phenomenon theoretically.

#### 1.4.2. Ion Exchange Capacity

IEC is a quantity of the capacity of an insoluble substance to endure ions displacement with formerly attached and lightly encapsulated into its architecture by oppositely charged ions existing in the adjacent solution. IEC was calculated by a back titration method with the following formula [50]:(2)$$\mathrm{IEC}\;\left(\mathrm{meq}.g^{-1}\right)=\frac{\mathrm{Titre}\;\mathrm{value}\;\times \;\mathrm{Normality}\;\mathrm{of}\;\mathrm{tirant}}{\mathrm{Membrane}\;\mathrm{weight}\;\left(\mathrm{dry}\right)}$$

#### 1.4.3. Ionic Conductivity

The ionic conductivity of the IEM was measured by AC impedance spectroscopy. Prior to the testing, the membranes (IEMs) of various forms were fully hydrated overnight in DI water. The measuring device with IEM was positioned in DI water to maintain the relative humidity (RH) at 100% throughout the experiment. Membrane resistance was measured from the difference in the resistance between the blank cell and the one with IEM separates the counter electrode and working electrode compartment and is converted into ionic conductivity values using the below formula [51]:(3)$$\mathrm{Ionic}\;\mathrm{conductivity}\;\left({S\;\mathrm{cm}}^{-1}\right)=\frac{L}{R\;\times \;A}$$
where R is resistance of IEM (ohm); L is width of IEM (cm); A is area of IEM (cm^2^).

#### 1.4.4. Methanol Permeability

The methanol permeability (p) is studied at ambient temperature using a two-portion diffusion cell comprising of a collector (C) and a reservoir (R). C and R were separated by the investigated IEM, occupied with DI water, methanol, respectively. Both the portions are stirred continuously during the permeability test. The methanol permeability is calculated from the time versus concentration curve of the methanol collector slope values according to the following equation [52].
(4)$$p\;\left({\mathrm{cm}}^{2}s^{-1}\right)=\frac{m\;\times {\;V}_{C\;}\times \;d}{A\;\times {\;C}_{R}}$$
where m represents the linear plot slope; V_C_ signifies the methanol solution volume in the C; A and d illustrate the area and thickness of the IEM; C_R_ is the methanol concentration in the tank.

#### 1.4.5. Selectivity Ratio

Especially for DMFCs, the IEM must have two significant characteristics. The proton/hydroxide ionic conductivity should be maximal and have minimal methanol diffusion. Therefore, the higher the ratio of ionic conductivity to methanol permeability (termed as selectivity ratio), the better the IEM performance of the DMFC. This selectivity ratio indicates the performance of the IEM [53].

#### 1.4.6. Oxidative Stability

Oxidative resistance is studied by Fenton’s test in terms of weight loss over a period. In Fenton’s reagent, degradation of the polymer is caused by free radicals attacking the electrophilic sites, leading to weight loss.

## 2. Polysulfone

Polysulfone is a commercially existing aromatic polymer. The relentless attention of the membrane researchers for PSU is because of its outstanding properties [54], such as soluble tendency in a wide range of solvents (dimethylformamide, dimethyl sulfoxide, halogen derivative, dimethyl acetamide, halogen derivatives), excellent film forming capacity, withstanding in high temperatures, wide range of operating pH, outstanding mechanical strength, and reasonable reactivity in aromatic electrophilic substitution reactions (acylation, chloromethylation, nitration, sulfonation, etc.) [55]. The chemical structure of polysulfone is shown in Figure 4. In this present review, we have seen the recent developments of the polysulfone-based membrane and its composites for PEMFCs, DMFCs, and AMFCs applications.

### 2.1. Membranes Derived from Polysulfone and Its Composites for PEMFCs Application

PEMFC technology has evolved quickly over the past 2 decades, with many advantages over traditional energy storage devices, such as batteries and internal combustion engines. PEMFCs are more energy-efficient related with diesel/gas engines. They also produce no hazardous by-products [56,57,58]. However, the practical feasibility of this technology is highly dependent on the PEM and its characteristics [59,60]. Hence, PEM is a crucial component in PEMFCs devices. The outcome of PEM depends not only on excellent mechanical and thermal resistance but also on the other characteristics, such as film-forming capacity, excellent proton conductivity, and reduced methanol cross-over [61,62]. Recently, nano fillers have been widely explored to adjust polymeric membranes to enhance the outcome of PEMs. These enhancements are reached by introducing a nonstop proton transfer path in the polymer environment and enhanced mechanical/ thermal characteristics of the polymer [63,64].

Metal-organic frameworks (MOFs) as carriers for proton-conducting material have received remarkable attraction from many experimental scientists owing to their high surface area compared to usual filler materials that permits encapsulation of proton transfer material [65,66]. For example, Leila Ahmadian-Alam and Hossein Mahdavi reported a ternary composite membrane composed of MOF and sulfonic acid functionalized silica (MOF/SO_3_H-*f*-Si) nanoparticles with polysulfone for PEMFCs [67]. The implanting of MOF/SO_3_H-*f*-Si nanoparticles on sulfonated PSU ensued in substantial enhancement of the thermal and mechanical properties of the composite membrane. The ion conductivity and transport properties of the composite membrane were increased to 0.017 S cm^−1^ by adding only 5% of MOF/SO_3_H-*f*-Si nanoparticles. Furthermore, the nanocomposite exhibited a supreme power density (PD) of 40.80 mW cm^−2^. Nor Azureen Mohamad et al. described cross-linked highly sulfonated polyphenylene sulfone (SPPSU) membranes comprised of carbon nanodots (CNDs) as a PEM for PEMFCs application [68]. The cross-linked membrane was prepared by pyrolysis at 453 K, where cross-linking occurs between SPPSU and CNDs. The prepared cross-linked composite membrane showed the maximum ionic conductivity of 56.3 mS cm^−1^. Further, the authors demonstrated that the CNDs encapsulation into SPPSU membrane by pyrolysis treatment displayed a high ionic conductivity with superior dimensional stability. Recently, Balappa B. Munavalli and Mahadevappa Y. Kariduraganavar have prepared PEM based composite membrane by two step methods. First, sulfanilic acid (H_2_N-C_6_H_4_-SO_3_H) functionalized poly(1,4-phenylene ether ether sulfone) (SPEESSA) was synthesized. Then, different weight percentages of -SO_3_H functionalized zeolites have been incorporated into the prepared composite membrane [69]. The composite membranes, Na-ZSM-5 zeolite, Na-β zeolite, and Na-Mordenite zeolite, exhibited the ionic conductivities of 102, 112, and 124 mS cm^−1^, respectively. Furthermore, the composite membrane with 8 weight% Na-ZSM-5 zeolite, Na-Beta zeolite, and Na-Mordenite zeolite exhibited outstanding PD of 0.37, 2.042, and 0.45 W cm^−2^, respectively, in H_2_/O_2_ fuel cells. In addition, the obtained PEMFCs results were much better than the commercially existing Nafion^®^ 117 membranes. Jinzhao Li et al. reported graphene oxide-based nanoscale ionic materials (NIMs-GO) by sulfonation with 3-(trihydroxysilyl)-1-propanesulfonic acid (SIT) and consequent neutralization with amino-terminated polyoxypropylene (PO)-polyoxyethylene (EO) block co-polymer [70]. The schematic illustration of the NIMs-GO synthesis is depicted in Figure 5(A1). Transmission electron microscopy (TEM) was employed to analyze the morphology of the prepared GO, SIT-GO, and NIMs-GO (Figure 5a–f). Despite sulfonation by SIT, GO nanosheets exhibit a wrinkled and folded configuration, an intrinsic property of GO due to their large surface area and intramolecular attraction. Remarkably, the NIMs-GO exhibited greatly stretched features (Figure 5e,f), after being ion-exchanged with M2070. The authors stated that the change in morphology confirms that the M2070 has been ionically bonded to the GO surface via -SO_3_H/-NH_2_ interactions. The resulting NIMs-GO with acid-base pairs and hygroscopic EO units were incorporated into sulfonated polysulfone (SPSF) to fabricate nanocomposite membranes. The water uptake and retention ability of the SPSF/NIMs-GO nanocomposite membranes were enhanced due to the hydrophilic EO units of NIMs-GO. Furthermore, the maximum PD of 167.6 mW cm^−2^ was attained for SPSF/NIMs-GO-3 at 60 °C/100% RH, which is higher than that of Nafion^®^ 117 and the pristine SPSF membrane (Figure 5(B1)). When the relative humidity drops to 50% (Figure 5(B2)), the maximum PD of 33.3, 17, and 23.2% decreases by SPSF, SPSF/NIMs-GO-3, and Nafion^®^ 117, respectively. All these results are due to the increased H^+^ conductivity of the fuel cell in both hydrated and low relative humidity conditions.

Recently, Cataldo Simari et al. synthesized sulfonated polysulfone (SPSF)/layered double hydroxide (LDH) nanocomposite membranes with various weight percentage filler content by an easiest solution intercalation method to replace Nafion^®^ electrolyte in PEMFCs applications [71]. The comprehensive exfoliation and nano dispersion of the LDH platelets into the polymer improve the thermomechanical resistance, water retention capability, and dimensional stability of the electrolyte membranes. The photographic images of the prepared PEMs were depicted in Figure 6a. All membranes except the sPSU-LDH_4_ membrane are transparent. In addition, no inorganic particles are noticed in both the sPSU-LDH_2_ and sPSU-LDH_3_ membranes illustrating no agglomeration. The power density and polarization curves are shown in Figure 6b,c. The maximum PD of 204.5 mW cm^−2^ at 110 °C/25% RH was achieved for the sPSU-LDH_3_ composite membrane, which is double the value achieved by the Nafion^®^ membrane. Such a superficial performance was attributed by the establishment of extremely interconnected ion pathways encouraging an efficient Ghrotthus-type mechanism for the H^+^ passage even in dehydrated environments.

Ting Pan et al. described novel composite membrane from functionalized PSU with high sulfonic acid groups, N,N-bis (sulfopropyl)aminyl-4-phenyl polysulfone (PSF-N-C_3_H_6_SO_3_H) and O,O’-bis(sulfopropyl)resorcinol-5-yl-4-phenyl polysulfone (PSF-O-C_3_H_6_SO_3_H) [72]. The above polymers prepared by grafting amino phenyl group and dimethoxy phenyl groups to the polymer backbone through bromination of PSU followed by Suzuki cross-coupling reaction, and the introduction of the sulfopropyl groups through sulfone ring-opening reaction. Furthermore, the prepared composite membrane exhibited the highest proton conductivity of 46.66 mS cm^−1^ at 95 °C/90% RH. In addition, the prepared membrane exhibited an adequate swelling ratio and water uptake and reduced methanol cross-over. The outstanding presentation of the composite membrane is due to the phase separation between the hydrophobic and hydrophilic subphases and the establishment of the hydrogen-bonding network in the hydrophilic subphase. Very recently, Berlina Maria Mahimai et al. prepared a series of nanocomposites from PSF, SPANI (sulfonated polyaniline), and Nb_2_O_5_ (niobium pentoxide) by the solution casting method [73]. The composite membrane with 10 wt% Nb_2_O_5_/PSF/SPANI displayed the highest ionic conductivity of 0.0674 S cm^−1^. Furthermore, the authors demonstrated that incorporating Nb_2_O_5_ into virgin PSF enhanced the proton conductivity and improved the thermal and oxidative stability.

In order to find different polymer electrolyte materials other than Nafion^TM^, polymeric membranes functionalized with H_3_PO_3_ (phosphonic acid) groups have encouraged much research consideration owing to their enhanced ionic conductivity at high temperature under dehydration environment ascribing to the self-ionization of H_3_PO_3_ groups within an infused hydrogen-bonding network [74,75]. When compared with SO_3_H and COOH, the H_3_PO_3_ group has moderate acidity and low water solubility and swelling ability, so it has a high ability for hydrogen bonding [76]. Furthermore, the bond that exists in phosphonic acid (-C-P-) is more thermally and electrochemically stable than the sulfonic acid (-C-S-) bond and carboxylic acid (-C-C-) bond, and therefore, more appropriate for PEMFCs application [77,78]. For example, Lesi Yu et al. reported proton-conducting composite membrane from SPSF and polysulfone grafted (phosphonated polystyrene) (SPSF/PPSF) through controlled atom transfer radical polymerization (ATRP) for PEMFCs application [79]. The supreme ionic conductivity of 0.01723 S cm^−1^ at 95 °C/90% RH was achieved. Furthermore, the SPSF/PPSF membrane exhibited promising thermal stability, adequate swelling ratio, and water uptake, notably enhanced mechanical stability. In addition, the permeability of methanol decreased from 5.74 × 10^−8^ cm^2^ s^−1^ for PPSF to 0.96 × 10^−8^ cm^2^ s^−1^ for the composite membrane.

PEMFCs operating at high temperatures (HT-PEMFCs) have received considerable attraction owing to their improved electrode reaction kinetics and simplified humidification and thermal management [80,81]. In the HT-PEMFC devices, the PEM is a vital element for carrying H^+^ (protons) and allocating fuel and oxygen. Hence, HT-PEMs necessitate both good ionic conductivity and adequate mechanical stability. There have been incredible efforts to progress HT-PEMs with high proton transport capacity at higher temperatures (120–300 °C). Recently, Hongying Tang et al. have prepared phosphate poly(phenylene sulfone) (P-PPSU) by post-phosphonylation of brominated poly(phenylene sulfone) (Br-PPSU), followed by acidification [82]. In addition, the prepared P-PPSU material can act as a binder material in the catalyst layer to decrease the decay of operating performance of HT-PEMFC operations. The ionic conductivity of P-PPSU membrane at a high temperature without extra humidification is only 0.30 mS cm^−1^ at 160 °C, the PD of 242 mW cm^−2^ is attained in fuel cell operation at 160 °C. The obtained values are low when compared with Nafion binder material; however, the excellent stability of 200 h is noticed in FCs worked at 160 °C with P-PPSU polymer binder with no noteworthy decrease in the fuel cell evaluation. Hence, the authors demonstrated that the prepared P-PSSU is a viable candidate as a binder material in the catalyst layer for extremely robust HT-PEMFCs. Jujia Zhang et al. have also prepared 2,4,6-tri(dimethylaminomethyl)-phenol (TDAP) with three tertiary amine groups that were grafted to PSF (TDAP-PSF) to attain higher phosphoric acid uptake at lower grafting degree from HT-PEMFCs [83]. Furthermore, the single cell reaches the PD of 453 mW cm^−2^ and has excellent stability without exterior humidification. Huijuan Bai et al. also described a new strategy for grafting poly(1-vinylimidazole) with phosphoric acid doping sites on the PSF backbone via ATRP [84]. The authors demonstrated that the high H^+^ conductivity is attained due to the establishment of micro-phase separated structures, and mechanical properties are maintained due to the decreased plasticizing effect produced by the separation of phosphoric acid adsorption sites and the polymer backbone. The obtained phosphoric acid incorporated membranes have outstanding ionic conductivity of 127 mS cm^−1^ at 160 °C and excellent tensile strength of 7.94 MPa. On the other hand, the single H_2_/O_2_ fuel cell performance with the optimized membrane is inspiring, achieving a peak PD of 559 mW cm^−2^ at 160 °C. Table 2 summarized the preliminary characteristics of various proton-conducting polysulfone-based composite membranes along with their fuel cell evaluations.

In summary, the potential of sulfonated polysulfone and its composites, phosphonated polysulfone, and several grafted polymers of sulfonated polysulfone for low and high-temperature polymer electrolyte membranes for PEMFC has been discussed. In general, the incorporation of nano sized inorganic filler or metal organic frameworks or zeolites has enhanced a phenomenal result in both the mechanical characteristics and ion conducting properties.

### 2.2. Polysulfone and Its Composites for DMFCs

DMFCs provide numerous distinct advantages associated with reasonable working temperatures, easy handling and storage of liquid fuel (methanol), offering power in the utmost effective way. Furthermore, there is no need to recharge the DMFC because liquid fuel can be delivered directly to the anode, and electricity can be produced immediately. Significantly, it might be the major energy basis for portable electronic instruments and automobiles with no toxic gases associated with combustion engines [85,86,87]. Sulfonated polysulfone (SPSU) exhibited exceptional mechanical strength and extraordinary methanol resistance (even at 100% sulfonation), illustrating its tremendous potential for the fabrication of novel polymeric membranes used in DMFC technology. Nevertheless, the very low ionic conductivity of SPSU still remains one of the most serious drawbacks. A favorable and cost-effective method to report this issue is to: (i) blend SPSU with other polymers. In practice, this approach is commonly applied in order to modify the characteristics of a virgin macromolecule, attaining superior properties in the resulting blended materials [88,89,90,91]. (ii) The preparation of composite membranes by dispersion of inorganic fillers including silica (SiO2) [92], titania (TiO2) [93], zeolites [94], and heteropoly acids inside the polymer matrix have been demonstrated to satisfactorily enhance the ionic conductivity of the resulting electrolyte without sacrificing its mechanical resistance [95,96]; and (iii) the introduction of functionalized 2D-layered materials (example, graphene oxide, smectite clay, layered double hydroxides (LDHs), and siliceous layered materials) effectively lowers the methanol permeability in Nafion-based membranes and simultaneously improves their proton conductivity, water retention capacity, and thermo-mechanical resistance [97,98,99,100]. Among these inorganic fillers, LDHs have recently gained more attention, a class of nanostructured materials belonging to the anionic clay family, with unique physicochemical properties [101,102,103,104]. For instance, E. Lufrano et al. described the incorporation of hygroscopic LDH particles into SPSU for DMFCs [105]. The substantial enhancement in the water and methanol absorption and dimensional stability of the SPSU/LDH composite membrane was observed when compared with both pristine SPSU and Nafion^®^ 212 membranes. Furthermore, the fabricated single DMFC achieved the remarkable PD of 150 mW cm^−2^ at 80 °C at higher methanol concentration (5 M methanol) solution. Xianlin Xu et al. reported bio-inspired amino acid-functionalized cellulose whiskers impregnated SPSU as PEM for DMFCs [106]. The maximum ionic conductivity of 0.234 S cm^−1^ at 80 °C achieved for 10 wt% L-Serine-functionalized cellulose whiskers. In addition, enhanced water uptake and reduced methanol cross-over were observed. Therefore, the composition of filler and mixed matrix display outstanding characteristics, and H^+^ conducting mixed-matrix membranes are promising materials in DMFCs. Adnan Ozden et al. prepared SPSU/zirconium hydrogen phosphate (ZrP) composite membranes with different degrees of sulfonation (20, 35, and 42%) and a uniform weight percentage of ZrP (2.5%) to alleviate the practical tasks related to the usage of traditional Nafion^®^ membranes in DMFCs [107]. The SPSU/ZrP-42 composite membrane exhibited a maximum OCV of 0.75 V and PD of 119 mW cm^−2^ at 80 °C. Nattinee Krathumkhet et al. synthesized composite membrane from sulfonated ZSM-5 zeolite and SPSU by solution casting method [108]. First, sulfonated ZSM-5 zeolite was synthesized by an organo-functionalization method using poly(2-acrylamido-2-methylpropanesulfonic acid). Then, SPSU was prepared by the conventional method. The composite membrane, ZSM-5/SPSU, significantly enhanced the ionic conductivity, water uptake, methanol cross-over, and IEC relative to the pristine SPSU membrane. Recently, C. Simari et al. reported blended electrolyte membranes comprised of SPSU and SPEEK (SPSU/SPEEK) with two different ratios, 50/50 and 25/75, through a facile and modest solution casting method for DMFC applications [109]. The fabricated blend membrane showed enhancement of the proton transport along with the reduced methanol cross-over which is one of the essential criteria for DMFC operation. Furthermore, the DMFC performance with 25/75 blend membrane showed a PD of 130 mW cm^−2^ at 353 K in 4 M methanol. Faizah Altaf et al. also prepared sulfonated polysulfone (SPSU) based composite PEM filled with polydopamine (PD) anchored carbon nanotubes (PD-CNTs) by phase inversion methodology with varying the filler (PCSPSU) [110] and the detailed reaction protocol was given in Figure 7. The composite membrane, 0.5 weight% PD-CNTs, displayed a 43% rise in ionic conductivity compared to the original SPSU membrane, increasing from 0.085 S cm^−1^ for pristine to 0.1216 S cm^−1^ for the composite membrane at 80 °C. The prepared composite membrane also exhibited a remarkable 75% reduction in methanol permeability (5.68 × 10^−7^ cm^2^ s^−1^) compared to recast Nafion^®^ 117 membranes (23.00 × 10^−7^ cm^2^ s^−1^). The obtained outcomes suggested that the PD functionalized CNTs based PEMs as a potential candidate for DMFCs.

In summary, SPSU membranes-based composite membranes were widely used as PEM for DMFCs to enhance its ionic conductivity, methanol cross-over, and single cell performance. Nevertheless, as previously discussed, in the performances of SPSU composites, blends, and LDH based SPSUs, many inconsistencies with the experimental results in relation to ionic conductivity, water uptake, and so on are perceptible. Each method used to improve the performance of composite and or blend SPSU based membranes offers benefits and drawbacks. Table 3 consists of SPSU and its composites for DMFCs application.

### 2.3. Alkaline Based Polysulfone and Its Composites for AMFCs

Recently, the progress of AMFCs has improved significantly, primarily due to the advantages of the existence of these systems over the widely known PEMFCs. The alkaline medium produced by AEM in the fuel cell favors electrode kinetics [111] and subsequently avoids the usage of expensive and noble metal catalysts. Hence, it is possible to use non-precious metals (cobalt, nickel and aluminium) [112], thereby reducing the cost of the system [113]. Nieves Urena et al. reported on amphiphilic semi-interpenetrating polymer networks for AEMFC applications with three dissimilar ionic groups, namely, tetramethyl ammonium, 1-methylimidazolium, and 1,2-dimethylimidazolium and cross-linked with N,N,N’,N’-tetramethylethylenediamine (TMEDA) [114]. The resulting membrane exhibits the following characteristic: (i) at low temperatures (lower than 100 °C) has high thermal stability, (ii) lower water uptake at ambient temperature, (iii) acceptable hydroxyl ion conductivity, (iv) outstanding chemical stability, (v) excellent dimensional stability because of the inferior water uptake. Furthermore, the membrane showed excellent alkaline stability. Recently, Yang Bai et al. prepared quaternized polysulfone-based AEMS cross-linked with rGO (CQPSU-X-rGO) functionalized with different chain length small molecules [115]. Especially, the functionalized CQPSU-X-rGO showed improved ionic conductivity and chemical stability. The maximum ionic conductivity of 0.140 S cm^−1^ at 80 °C was achieved for rGO cross-linked AEMS. Tiantian Li et al. synthesized PSU based anion exchange membrane via Friedel-Crafts alkylation method contains pendant imidazolium functionalized side chain to avoid conventional carcinogenic chloromethylation. It does not require any special functional groups on the polymeric materials, which is the main advantage compared with other mentioned chloromethylation-free routes in the literature [116]. Furthermore, the membranes synthesized in this methodology displayed excellent ionic conductivity and swelling ratio along with good mechanical, thermal, and alkaline stabilities. Very recently, Lingling Ma et al. synthesized a series of AEMs modified with bulky rigid -cyclodextrin (CD) and long flexible multiple quaternary ammonium (MQ) membrane for AMFC applications [117]. The resulting AEM with a relatively low IEC of 1.50 meq. g^−1^ exhibits a good ionic conductivity of 112.4 mS cm^−1^ at 80 °C, whereas its counterpart without CD modification shows 83.0 mS cm^−1^ despite a similar ion exchange capacity (1.60 meq. g^−1^). This is because large CD units can impart a high free volume to the membrane, dropping the ion transfer resistance, while the hydrophilicity of the external surface of the CD can promote the formation of ion transport channels across the long flexible MQ cross-links. The fabricated H_2_/O_2_ FC provides a maximum PD of 288 mW cm^−1^ at 60 °C. Mona Iravaninia et al. prepared AEM from polysulfone membrane by a conventional three-step method, chloromethylation, amination, alkalization with functionalized trimethylamine and N,N,N’,N’-tetramethyl-1.6-hexanediamine [118]. The prepared membrane exhibited ionic conductivity of 2–42 mS cm^−1^ at 25–80 °C in different RH. The IECs, anion transport numbers, and hydration numbers were within the range of 1.6-2.1 meq. g^−1^, 0.95–0.98 and 9–16, respectively. Furthermore, the single H_2_/O_2_ fuel cell showed a OCV of 1.05 V and a maximum PD of 110 mW cm^−2^ at 60 °C. Yang Bai et al. proposed a facile strategy to construct rGO stable cross-linked PSU-based AEMs with enhanced properties [119]. The cross-linked AEMS can constrict the internal packing structure and improve alkaline stability, ion conductivity, and oxidative stability. The rGO cross-linked AEM showed higher ionic conductivity of 117.7 mS cm^−1^ at 80 °C. Wan Liu et al. derived AEM from QPSU and exfoliated LDH for fuel cell applications [120]. The composite membrane comprising 5% LDH sheets showed good performance, displaying an ionic conductivity of 0.0235 S cm^−1^ at 60 °C. Yuliang Jiang et al. reported a series of PSU-based AEMs with cross-linker, 4, 4′-trimethyenedipiperidine (TMDP) [121]. The cross-linked aminated polysulfone (CAPSF) displayed supreme alkaline stability compared with non-crosslinked aminated polysulfone (APSF) in 1 M KOH for 15 days at 333 K. Furthermore, the CAPSF exhibits better dimensional stability as compared with the non-cross-linked APSF membrane owing to the compact interconnected architecture formation. From the above results, the authors concluded that the prepared crosslinked AEM is a potential candidate for AMFCs. Maria Teresa Perez-Prior et al. prepared crosslinked polysulfone AEMs using 1,4-diazabicyclo [2,2,2] octane (DABCO) as cross-liner [122]. The obtained results revealed that the cross-linked membranes displayed exceptional thermal stability, improved water uptake and dimensional stability as compared with non-cross-linked AEM. Prerana Sharma et al. described a novel strategy to synthesize alkaline membrane of chloromethylated polysulfone using cross-linker, 4,4′(3,3′-bis(chloromethyl)-[1,1′-bipheny]-4,4-diyl)bis(oxy))dianiline) (BCBD) [123]. The detailed reaction pathway of cross-linked quaternary polysulfone (CR-QPS) membrane is shown in Figure 8. The cross-linked membrane performed well in AMFCs and exhibited maximum OCV of 0.813 V and PD of 103.6 mW cm^−2^ at 260 mA cm^−2^.

P. F. Msomi et al. reported a sequence of AEM comprised of poly(2,6-dimethyl-1,4-phenylene) (PPO) and PSF blended with titania (QPPO/PSF/TiO_2_) [124]. The swelling ratio, ionic conductivity, water uptake, and IEC of the composite were enhanced by multiplying the titania filler content. Furthermore, the QPSU/PSF/2% TiO_2_ displayed a supreme PD of 118 mW cm^−2^ at 60 °C with excellent membrane stability over 60 h. K. Rambabu et al. described imidazolium functionalized PSF membranes modified with zirconia (Im-PSF/ZrO_2_) by solution casting method for AMFC applications [125]. The enhanced water absorption, IEC (2.84 meq. g^−1^), hydroxyl ion conductivity (80.2 mS cm^−1^ at 50 °C), and thermal resistance achieved for Im-PSF/ZrO_2_ composite membrane as compared with pristine Im-PSF, which confirms the strong adhesion and property enhancement caused by zirconia. Furthermore, the composite membrane with Im-PSF/10% ZrO_2_ showed a maximum PD of 270 mW cm^−2^ with OCV of 1.04 C in H_2_/O_2_ fueled AMFCs.

In summary, prominent developments have been made for the use of quaternized polysulfone with AEM in alkaline membrane fuel cells with respect to thermal, electrochemical, mechanical stability, and hydroxyl ion conductivity. Furthermore, virtuous advancement has been achieved regarding the impregnation of various inorganic filler or ionic liquids or polymer blend into various polymeric assemblies where the resultant AEMs accomplished rational performance when tested in AMFCs.

## 3. Conclusions and Future Perspectives

The emerging fuel cell market is a strong driving force for the scientific community to achieve new, affordable, and high-performance membrane materials. The present review deals with the recent advancements of polysulfone-based proton exchange membrane/anion exchange membrane for PEMFCs, DMFCs, and AMFCs application. Polysulfone-derived PEM/AEM and its composites are exploited a crucial role in the fuel cell applications as evidenced by the ample literature that is available. For PEMFCs/DMFCs, sulfonated polysulfone and its composites with inorganic fillers, layered double hydroxides, metal-organic frameworks have been investigated in this present review. Specifically, water uptake, ionic (H^+^) conductivity, methanol permeability, alkaline stability, and the performance of fuel cell substantially enhanced as compared with pristine sulfonated polysulfone. Furthermore, many polymer electrolyte membranes reported in this review showed a better fuel cell performance and reduced methanol crossover compared with commercially available Nafion membranes in both PEMFCs and DMFCs operation. However, still Nafion membranes were used in the industrial sector and transport vehicles. Therefore, the commercialization of the PEMs is the utmost priority to every researcher in the membrane study to overcome Nafion membrane for PEMFCs/DMFCs applications.

As deliberated, the use of AEMs in electrochemical systems could potentially eliminate the common issues such as fuel crossover, confronted in PEMFCs. Additionally, the use of AEMs has several advantages, such as being used in alkaline environments, which enables the use of non-precious metal catalysts. Nevertheless, numerous problems need to be fixed such as poor ionic conductivity (which is accountable for poor voltage efficiency and ohmic losses), insufficient membrane stability in alkaline and oxidative atmospheres, and a lack of suitable alkaline ionomers, especially for AMFCs. Several conventional methods have been extensively studied to improve the ionic conductivity of AEMs. Recently, interpenetrating polymer network (IPN) and pore-enriched composite AEMs have efficiently imitated the Nafion-like morphology, where the hydrophobic polyolefin and the hydrophilic quaternized polymer moiety are well disconnected. As a result, a fabulous enhancement in the ionic conductivity could be attained. Inclusive data regarding the oxidative stability of AEMs can inspire further work towards the modification of existing materials or the development of new materials for AEMs. The development of AEMs based on PEEK, polybenzimidazole, and functional group chemistries based on imidazolium and guanidinium are still in the early stages. Therefore, the chemical stability of these AEMs can be studied in detail and their performance in electrochemical systems can be explored extensively.

Furthermore, the aminated/quaternized polysulfone blended with other polymers or the incorporation of inorganic fillers, such as silica, titania, zirconia, zeolites, metal-organic frameworks, etc., hinder the ionic conductivity and may reduce the chemical stability of the AEM. Despite their low alkaline stability, AEMs is still an important research field with a great outlook due to their outstanding advantages over PEMFCs. Therefore, there is an urgent need to progress novel AEMs that attain a high ionic conductivity and selectivity and exhibit outstanding chemical stability in alkaline conditions and high temperatures.

## Figures and Tables

**Figure 1 polymers-14-00300-f001:**
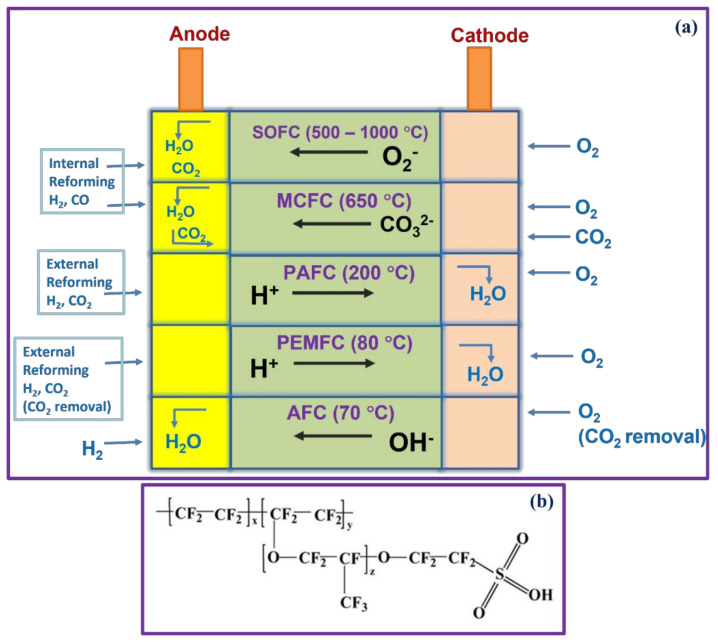
(**a**) Different types of fuel cell; (**b**) structure of PFSA.

**Figure 2 polymers-14-00300-f002:**
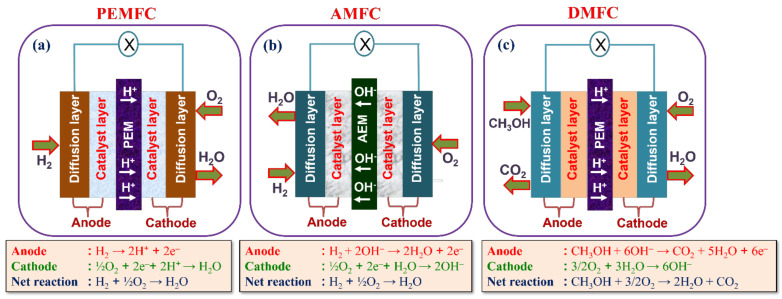
Schematic illustration of (**a**) PEMFC; (**b**) AMFC, and (**c**) DMFC with half-cell reaction.

**Figure 3 polymers-14-00300-f003:**
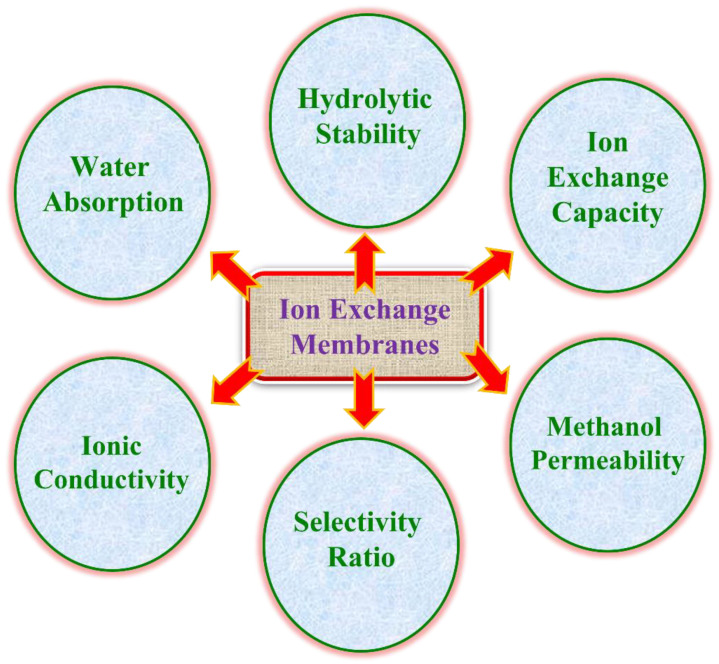
Characteristics of IEMs for FC applications.

**Figure 4 polymers-14-00300-f004:**
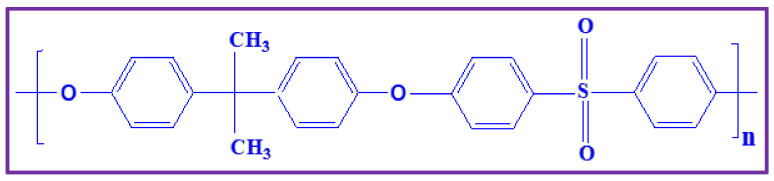
Chemical structure of PSU.

**Figure 5 polymers-14-00300-f005:**
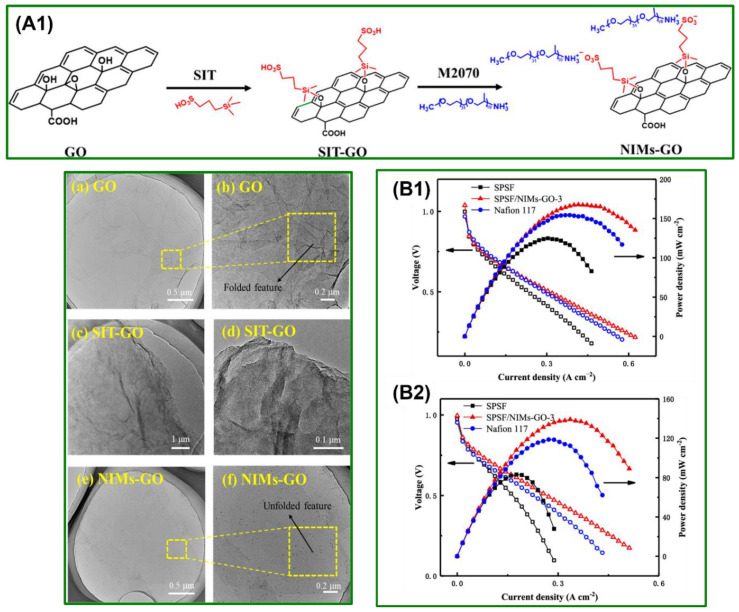
(**A1**) Reaction protocol of SIT-GO and NIMs-GO; (**a–f**) represents the TEM images of the prepared GOs; H_2_/O_2_ fuel cell performances at (**B1**) 60 °C/100% RH and (**B2**) 60 °C/50% RH. Reproduced with permission from [70]. Copyright 2019 American Chemical Society.

**Figure 6 polymers-14-00300-f006:**
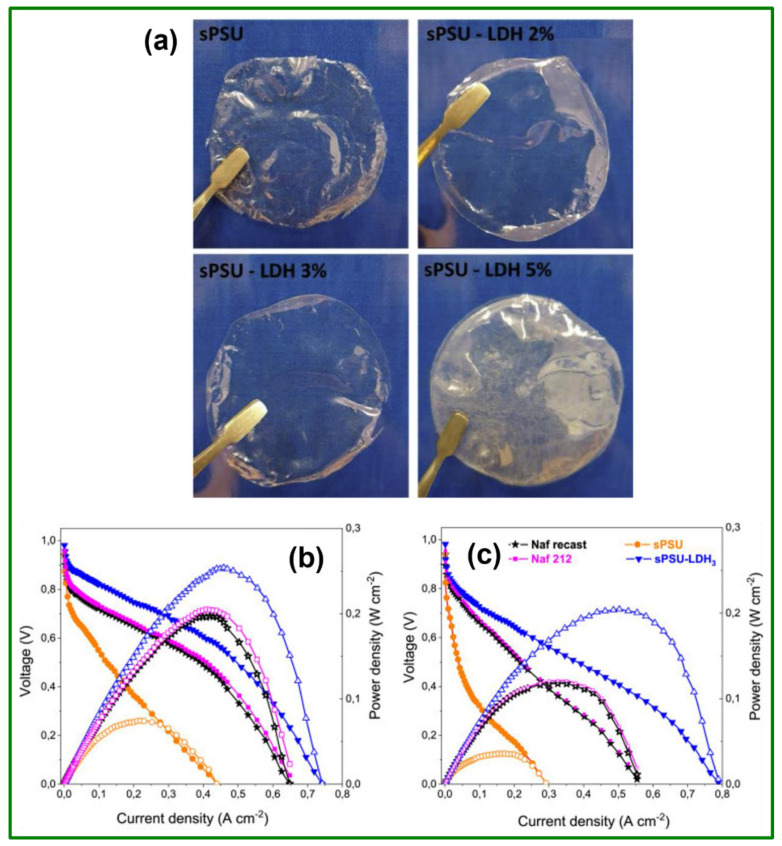
(**a**) Pictorial representation of the prepared membranes; cell voltage and PD plots of H_2_/O_2_ fuel cell at (**b**) 80 °C/30% RH and (**c**) 110 °C/25% RH. Reproduced with permission from [71]. Copyright 2020 Elsevier.

**Figure 7 polymers-14-00300-f007:**
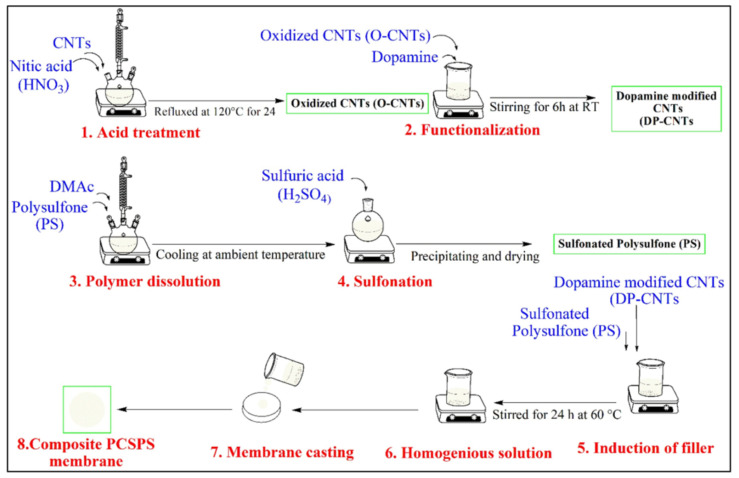
Reaction protocols entailed in the preparation of composite membrane, PCSPSU. Reproduced with permission from [110]. Copyright 2020 Elsevier.

**Figure 8 polymers-14-00300-f008:**
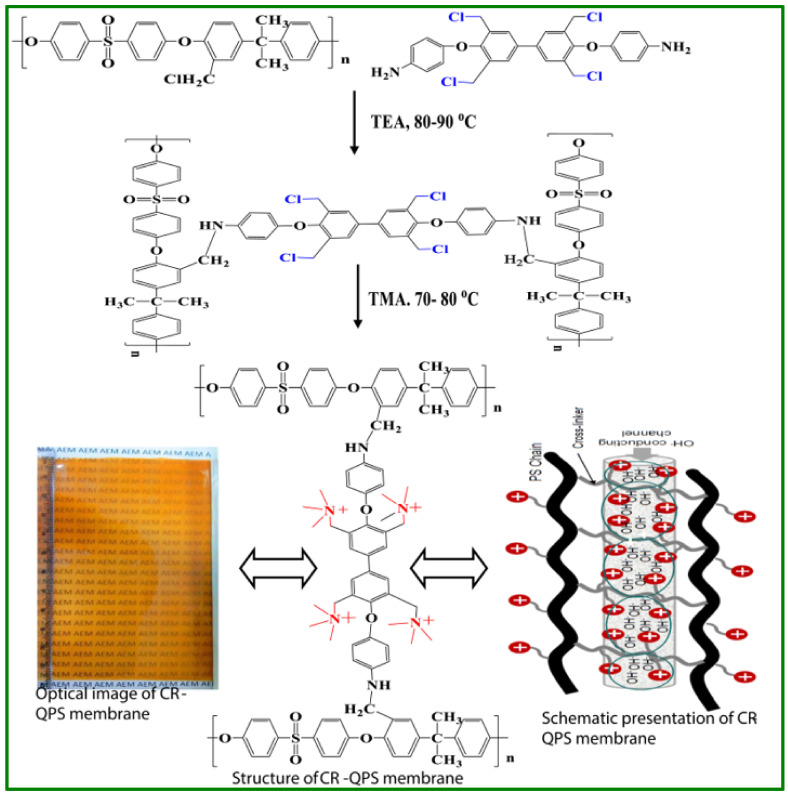
Schematic illustration for synthesis of CR-QPS AEM. Reproduced with the permission from [123]. Copyright 2020 Elsevier.

**Table 1 polymers-14-00300-t001:** Difference between cation and anion exchange membrane.

IEM	CEM	AEM
Counter ion	H^+^ conductive	OH^−^ conductive
Ion-exchange group	-SO_3_^−^; -PO_4_^−^; -CO_2_^−^	Quaternary ammonium cation, 1-methyl pyridinium
Features	High ionic conductivity, excellent ionomer solution	Non-noble metal catalyst can be used. Oxygen reduction reaction and methanol oxidation reaction are more facile.
Issues	High-cost materials, fuel crossover, chemical, and mechanical stability, practical lifetime	Low ionic conductivity, low thermostability, influence of CO_2_, durability, chemical, and mechanical stability

**Table 2 polymers-14-00300-t002:** Preliminary characteristics of various proton-conducting polysulfone based composite membranes along with their fuel cell evaluation.

Membrane	Membrane Characteristics						Fuel Cell Performance	Ref.
	WA (%)	IEC (meq. g^−1^)	Ionic Conductivity (S cm^−1^)	Methanol Permeability	Selectivity Ratio	Oxidative Stability		
PSF/MOF/Si nanocomposite	16.50	0.86	0.017 @ 70°C	---	---	---	OCV: 0.90 V; PD: 40.80 mW cm^−2^ @ 160 °C	[67]
Crosslinked CNDs-SPPSU	134	1.67	0.0563 @ 80 °C	---	---	---	OCV: 1.0224 V @ 100% RH	[68]
SPEESSA/sulfonic acid zeolite composite	29.12	3.189	0.124	---	---	---	OCV: 0.91 V; PD: 0.45 W cm^−2^ @ 1.1 A cm^−2^	[69]
SPSU/NIMs-GO composite	34.1	1.49	0.23 @ 75 °C	---	---	---	OCV: 1.038 V; PD: 167.6 mW cm^−2^ @ 60 °C	[70]
SPSU-LDH composite	31	1.49	0.0137 @ 120 °C	---	---	---	PD: 204.5 mW cm^−2^ @ 110 °C	[71]
PSF-N-C_3_H_6_SO_3_H/ PSF-O-C_3_H_6_SO_3_H	60	2.03	0.04666	2.65 × 10^−8^ cm^2^ s^−1^	---	94.12% residual mass remains at 80 °C for 1 h in Fenton’s solution	---	[72]
PSU/SPANI/Nb_2_O_5_ nanocomposite	17.6	1.50	0.0674	---	---	98.6% residual mass remains in Fenton’s solution	---	[73]
PSU-g-phosphonated polystyrene/SPSU composite	23.07	---	0.0172 @ 95 °C	0.96 × 10^−8^ cm^2^ s^−1^	---	>95% residual mass remains at 25 °C for 120 h in Fenton’s solution	---	[79]
Phosphonated PSU	6.6	2.75	0.0003 @ 160 °C	---	---	87.7% residual mass remains for 70 h in Fenton’s solution	---	[82]
PA doped TDAP-g-PSU	---	---	0.056 @ 160 °C	---	---	---	OCV: 0.92 V; PD: 453 mW cm^−2^ @ 150 °C	[83]
Poly(1-vinylimidazole)-g-PSU	220.3	---	0.127 @ 160 °C	---	---	---	OCV: 0.98 V; PD: 559 mW cm^−2^ @ 160 °C	[84]

**Table 3 polymers-14-00300-t003:** Sulfonate polysulfone and its composites for DMFCs.

Membrane	Membrane Characteristics						Fuel Cell Performance	Ref.
	WA (%)	IEC (meq.g^−1^)	Ionic Conductivity (S cm^−1^)	Methanol Permeability (cm^2^ s^−1^)	Selectivity Ratio (sS cm^−3^)	Oxidative Stability		
SPSU/LDH nanocomposite	29	1.49	0.102 @ 120 °C	116 mA cm^−2^	---	---	OCV: 0.82 V; PD: 150 mW cm^−2^ @ 80 °C in 5 M CH_3_OH	[105]
Amino-acid functionalized cellulose whiskers/SPSU	68	---	0.234 @ 80 °C	7.6 × 10^−7^	---	---	OCV: 0.73 V; PD: 73.757 mW cm^−2^ @ 60 °C in 2 M CH_3_OH	[106]
SPSU/ZrP	38	---	0.156 @ 80 °C	---	---	96.66% of weight retention after Fenton test	OCV: 0.75 V; PD: 119 mW cm^−2^ @ 80 °C	[107]
Sulfonated ZSM-5/SPSU	45.41	1.03	0.00965 @ RT	2.24 × 10^−6^	4309.03	---	---	[108]
SPSU/SPEEK	34	---	0.073 @ 120 °C	---	---	---	OCV: 0.81 V; PD: 130 mW cm^−2^ @ 80 °C in 4 M CH_3_OH	[109]
PD-CNT/SPSU composite	32	---	0.1216 @ 80 °C	5.68 × 10^−7^	---	---	---	[110]

## Data Availability

The data presented in this study are available on request from the corresponding author.

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
