# Peer review of "Recent Advancements in Polysulfone Based Membranes for Fuel Cell (PEMFCs, DMFCs and AMFCs) Applications: A Critical Review"

_polymers, 2022, doi:10.3390/polym14020300_

Round 1
Reviewer 1 Report
In this manuscript, Polysulfone (PSU)-based electrolyte membranes with excellent characteristics as a promising substitute membrane in fuel cell systems are reviewed, including the applications in PEMFCs, AMFCs, and DMFCs. Various fillers incorporated composite membrane is the main focus, and the key issues associated with enhancing the ionic conductivity and chemical stability have been elucidated as well. Furthermore, this review addresses the current tasks and forthcoming the future directions of PEM/AEMs for PEMFCs, DMFCs, AMFCs briefly.
I consider the content of this manuscript will definitely meet the reading interests of the readers of the Polymers journal. However, the discussion is still slightly monotonous and confusing, and the introduction needs to be further improved.
Therefore, I suggest giving a major revision and the authors need to clarify some issues or supply some more data to enrich the content.
More detailed comments can be found in the PDF document attached.

Author Response
Answer to the Reviewer#1 Comments
Journal: Polymers
Manuscript ID: polymers-1525713
We thank the editors and reviewers for their valuable comments in respect of this manuscript. As per their comments and suggestions, the manuscript is thoroughly checked, appropriately modified, corrected the mistakes and new experimental results are added and discussed. The answers to the comments of the reviewers, point by point, are given as follows, and some of the answers are incorporated in the revised manuscript as well.
Reviewer#1
In this manuscript, Polysulfone (PSU)-based electrolyte membranes with excellent characteristics as a promising substitute membrane in fuel cell systems are reviewed, including the applications in PEMFCs, AMFCs, and DMFCs. Various fillers incorporated composite membrane is the main focus, and the key issues associated with enhancing the ionic conductivity and chemical stability have been elucidated as well. Furthermore, this review addresses the current tasks and forthcoming the future directions of PEM/AEMs for PEMFCs, DMFCs, AMFCs briefly.
I consider the content of this manuscript will definitely meet the reading interests of the readers of the Polymers journal. However, the discussion is still slightly monotonous and confusing, and the introduction needs to be further improved.
Therefore, I suggest giving a major revision and the authors need to clarify some issues or supply some more data to enrich the content.
Question: 1. Abstract and Introduction
The title is incomplete, and the end of the title should be: ‘A Critical Review’.
Answer: We sincerely thank the reviewer for the kind notice and valuable suggestion. The authors submitted the manuscript in word format, not in a journal template. After submission, the journal office changed the format before sending it to reviewers. Thus, the formats are changed. While submitting the manuscript to Polymers Journal at that time itself, the title was ended with “A critical review”.
Question: Line 14, ‘Currently, Nafion (perfluoro sulfonic acid) membrane has been widely employed in the membrane industry in proton-exchange membrane fuel cells... suffers reduced proton conductivity at a higher temperature...’ Grammar problems need special attention, especially the wrong use of preposition collocation and the lack of definite articles.
Answer: We appreciate and comply with the referee’s comments. As per the reviewer's suggestion, the corrections mentioned above have been rectified in the revised manuscript. In addition, we strongly apologies for the mistake. The manuscript was thoroughly checked for grammatical errors/mistakes/typo errors, and it has been rectified in the revised manuscript.
Question: For the Keywords, ‘fillers’ and ‘inorganic-organic hybrid membrane’ should also be added to further attract a broader readership.
Answer: We appreciate and comply with the referee’s comments. As per the reviewer suggestion, the above mentioned keywords have been rectified in the keywords section of the revised manuscript.
Question: Line 33, ‘As a new energy technology, fuel cells have been shown to be highly efficient and have an excellent ability to convert conventional fossil fuel energies due to low or zero-emission [1-3].’ This explanation is correct, but it is still incomplete. Because of the lack of description of hydrogen-oxygen fuel cells and their combination with renewable energy. Converting some fossil fuels into electricity through fuel cells is only part of the application of fuel cells. More importantly, most renewable energy sources are intermittent, opening spatial and temporal gaps between the availability of the energy and its consumption by the end-users [Electrochimica Acta 309 (2019): 311-325].
Therefore, the electrolysis device can be used to electrolyze water with the electric energy generated by the excess renewable energy, collect the generated hydrogen and oxygen, and convert the chemical energy in hydrogen and oxygen into electric energy through the fuel cell when the renewable energy capacity is interrupted. In this way, the peak regulation of renewable energy is realized.
Answer: We appreciate and comply with the referee’s comments. The necessary corrections have been carried out in the revised manuscript.
Question: Part 1.1 Fuel cells, as an electrochemical energy conversion design, what is the advantages of fuel cells over other commonly used electrochemical energy storage systems, for example, lithium-ion battery and flow battery [Renewable and Sustainable Energy Reviews 89 (2018): 292-308; Journal of Power Sources 493 (2021): 229445]? A simple comparison should be made to give readers a more intuitive understanding, which can also highlight the importance of this review.
Answer: We appreciate and comply with the referee’s comments. As per the reviewer's suggestion, a simple comparison of battery and fuel cell has been discussed along with why fuel cells are superior to batteries in the revised manuscript.
Question: Line 35, ‘In addition, FCs are highly stable devices since they will not release toxins like CO2, NOx, and SOx.’ This description is also inaccurate. Does DMFC really not produce carbon dioxide, a greenhouse gas? Look at the reaction equation below. Otherwise, where does the carbon in methanol go? At the same time, I must point out that carbon dioxide is not a toxic gas, but a greenhouse gas. Carbon monoxide is a toxic gas.
Answer: The given comment is valid and accepted. The above sentence in the revised manuscript has been modified as “fuel cells are providing an inherently clean source of energy, with no adverse environmental impact during operation as the byproducts are simply heat and water”.
Question: Line 60, ‘Nafion is pre-dominantly employed because of its excellent proton conductivity and adequate chemical/mechanical characteristics; they are worked at temperatures below 90-100 C and high relative humidity [11,12].’ Here, which structure leads to its excellent proton conductivity and good chemical/mechanical properties should be explained better. Proton conductivity is due to the significant phase separation between hydrophilic and hydrophobic domains in Nafion when hydrated. And the chemical/mechanical stability is caused by the rigid structure of the PTFE backbone and strong C-F bond even in the side chains [Solid State Ionics 319 (2018): 110-116].
Answer: (i) We appreciate and comply with the referee’s comments. As per the reviewer's suggestion, the corrections mentioned above have been rectified in the revised manuscript. (ii) Also, the causes of proton conductivity, chemical/mechanical stability of perfluorinated sulfonic acid membranes have been incorporated in the revised manuscript.
Question: In addition, Line 63 ‘It contains a sulfonic acid group pendant to the polytetrafluoroethylene backbone’ is also not correct. See Figure 1b, it is clear that the sulfonic acid group is connected to the perfluoroethereal side chains of the Nafion, not the backbone. And the membrane is not possible to have proton conductivity only due to the hydrophilic phase (-SO3H) since there is no phase separation to form the ion transport channels.
Answer: The given comment is valid and accepted. The corrections mentioned above have been rectified in the revised manuscript.
Question: In Table 1, information about chemical stability, mechanical stability, and practical lifetime/durability is still missing, which is also very critical for long-term FC operations.
Answer: We appreciate and comply with the referee’s comments. As per the reviewer suggestion, the information mentioned above has been incorporated in the Table 1 of the revised manuscript.
Question: Line 130, ‘The prepared IEMs were subjected to the following preliminary characterization studies like...’
Answer: We appreciate and comply with the referee’s comments. As per the reviewer suggestion, the corrections mentioned above have been rectified in the revised manuscript.
Question: Line 138, for WU% calculation, how the Wwet and Wdry are obtained should be further explained. Generally, when measuring the Wwet, after immersion in water, the surface water traces should be removed before the measurement. And for Wdry, the membrane should be dried in airflow or in the oven at a certain temperature [see Section 2.5 of Solid State Ionics 319 (2018): 110-116].
Answer: The given comment is valid and accepted. The water uptake section has been further improved in the revised manuscript. In addition, the reviewer suggested reference has been incorporated in the revised manuscript. Please see reference [49].
Question: Line 147, ‘Prior to the testing, the IECs of H+/OH- form was fully hydrated overnight in DI water. ’ No, it should be the membranes (IEMs) of various forms were fully hydrated
overnight in water. IEC is the ion-exchange capacity, it cannot be immersed/hydrated but only be measured.
Answer: We appreciate and comply with the referee’s comments. As per the reviewer suggestion, the corrections mentioned above have been rectified in the revised manuscript.
Question: Line 150, ‘... the resistance between the blank cell and the one with IEM separates the counter electrode and working electrode compartment and is converted into ... ’
Answer: We appreciate and comply with the referee’s comments. As per the reviewer suggestion, the corrections mentioned above have been rectified in the revised manuscript.
Question: Line 172, ‘In Fenton’s reagent, degradation of the polymer is caused by free-radicals attacking on the electrophilic sites, leading to weight loss.’
Answer: We appreciate and comply with the referee’s comments. As per the reviewer suggestion, the corrections mentioned above have been rectified in the revised manuscript.
Question: 2. Polysulfone
Line 220, ‘Then, different weight percentages of -SO3H functionalized zeolites have been incorporated into the prepared composite membrane [69]. ’
Answer: We appreciate and comply with the referee’s comments. As per the reviewer suggestion, the corrections mentioned above have been rectified in the revised manuscript.
Question: Line 312, ‘In addition, the prepared P-PPSU material can act as a binder material in the catalyst layer to decrease...’
Answer: We appreciate and comply with the referee’s comments. As per the reviewer suggestion, the corrections mentioned above have been rectified in the revised manuscript.
Question: Line 334, ‘Table 2 summarized the preliminary characteristics of various proton-conducting polysulfone based composite membranes along with their fuel cell evaluations’. The same applies to Line 343.
Answer: We appreciate and comply with the referee’s comments. As per the reviewer suggestion, the corrections mentioned above have been rectified in the revised manuscript.
Question: Line 404, when compared the values of methanol permeability, it is better also to mention the thickness of each membrane. Generally, a thicker membrane will prevent methanol permeability more efficiently. By comparing two membranes with unknown thicknesses, the reduction of methanol permeability makes only partial sense, like a very thick membrane (even possesses very low methanol permeability) will lead to very high resistance and ohmic loss, and it is not suitable for practical applications.
Answer: We appreciate and comply with the referee’s comment. Giving due respect to reviewer comment, the authors didn’t mention the thickness of the Nafion 117 and prepared composite membrane. Therefore, we couldn’t be able to give the thickness of the membranes.
Question: Line 414, ‘many inconsistencies with the experimental results in relation to ionic conductivity, water uptake, and so on are perceptible. Each method used to improve...’
Answer: We appreciate and comply with the referee’s comments. As per the reviewer suggestion, the corrections mentioned above have been rectified in the revised manuscript.
Question: Line 425, ‘Hence, it is possible to use non-precious metals (silver, cobalt, and nickel) [112].’ This description is wrong. The shortlist of chemically noble metals comprises ruthenium (Ru), rhodium (Rh), palladium (Pd), osmium (Os), iridium (Ir), platinum (Pt), gold (Au) and silver (Ag). So silver is a noble/precious metal, and the example the authors made in the text is not correct.
Answer: The given comment is valid and accepted. As per the reviewer's suggestion, the silver has been removed in the revised manuscript.
Question: Line 445 and Line 277, ‘swelling ration’ should be ‘swelling ratio’.
Answer: We appreciate and comply with the referee’s comments. In addition, we strongly apologies for the mistake. The corrections mentioned above have been rectified in the revised manuscript.
Question: Line 488, ‘The ... and IEC of the composite were enhanced by multiplying the titania filler content... with excellent membrane stability over a 60 h.’
Answer: We appreciate and comply with the referee’s comments. As per the reviewer suggestion, the corrections mentioned above have been rectified in the revised manuscript.
Question: 3. Conclusions and future perspectives
Line 509, ‘Polysulfone derived PEM/AEM and its composites are exploited a crucial role in the fuel cell applications as evidenced by the ample literature that is available. ’
Answer: We appreciate and comply with the referee’s comments. As per the reviewer suggestion, the corrections mentioned above have been rectified in the revised manuscript.
Question: Line 527, ‘Despite their low alkaline stability, AEMs is still an important research field with a great outlook due to their outstanding advantages over PEMFCs. ’
Answer: We appreciate and comply with the referee’s comments. As per the reviewer suggestion, the corrections mentioned above have been rectified in the revised manuscript.
Question: Line 529, ‘Therefore, there is an urgent need to progress a novel AEMs that not only attain a high ionic conductivity and selectivity but also exhibit outstanding chemical stability’
Answer: We appreciate and comply with the referee’s comments. As per the reviewer suggestion, the corrections mentioned above have been rectified in the revised manuscript.

Reviewer 2 Report
I am not keen on the name Proton exchange membrane fuel cells (PEMFCs). I prefer Polymer Electrolyte Membrane fuel cell (PEMFC).
Nafion needs a TM - Nafion™
Perfluorinated sulfonic acid NafionTM has no space in name. And use PFSA, and use PFSA (not Nafion) for Fig 1 (b). It is not the correct structure for all Nafion.
And do not FC for fuel cell. And not use, LTFC or HTFC.
The “best-known proton conductor is PFSA, which consist of a hydrophobic 42 polytetrafluoroethylene (PTFE)” – this is correct but one should also refer to NafionTM
Typo in Fig 1 for OH-
Typo is missing ‘degree’ page 62
Page 62 – Most automotive PEMFC will operate at above 100 degrees C in higher pressure.
Page 82 typo (missing spaces in many locations)
Line 269 has off page para from ‘TAB’ is missing
OVERALL – the ‘Jargon’ must be set-up with to many typos and the ‘English’ must be improved.
Water in PFSA including Schroeder’s paradox; however, it contains a nonthermodynamic assumption of constant water activity (equal to 1) for water contents beyond equilibrium. This should be discussed for a few lines. The Schroeder’s paradox should discussed the implications of this new membrane material.
In ‘light vehicles’ fuel cells are pressurized (and above 100 degree C). This should be discussed in the background.
The chemistry structure should have subscripts.
For a review paper, this paper has too many typos, references are outdated, and Jargon is weak.
Page 351 – I disagree. The Anode use 99.995 hydrogen, so the ‘toxic’ is only a concern for the cathode. This discussion should be improved.
Line 512 - “PEMFCs/DMFCs, 511 sulfonated polysulfone and its composites with inorganic fillers, layered double hydrox- 512 ides, metal organic frameworks have been investigated in this present review.” This is correct, but this has been known for 20 years. The advance of AEM should be updated.
Too many references are very OLD (10 or 20+ old). Format for the references out poor presentation.
Even a Review Paper is outdated, and nothing is novel in the analysis. References are outdated.
Author Response
Answer to the Reviewer#2 Comments
Journal: Polymers
Manuscript ID: polymers-1525713
We thank the editors and reviewers for their valuable comments in respect of this manuscript. As per their comments and suggestions, the manuscript is thoroughly checked, appropriately modified, corrected the mistakes, and new experimental results are added and discussed. The answers to the comments of the reviewers, point by point, are given as follows, and some of the answers are incorporated in the revised manuscript as well.
Reviewer#2
Question: I am not keen on the name Proton exchange membrane fuel cells (PEMFCs). I prefer Polymer Electrolyte Membrane fuel cell (PEMFC).
Answer: Thank you very much for your valuable suggestion. Polymer Electrolyte Membrane fuel cells (PEMFC) were used in the revised manuscript instead of Proton exchange membrane fuel cells.
Question: Nafion needs a TM - Nafion™
Answer: We appreciate and comply with the referee’s comments. As per the reviewer suggestion, Nafion was replaced with NafionTM in the entire manuscript.
Question: Perfluorinated sulfonic acid NafionTM has no space in name. And use PFSA and use PFSA (not Nafion) for Fig 1 (b). It is not the correct structure for all Nafion.
Answer: We appreciate and comply with the referee’s comment. As per the reviewer suggestion, the given comment is rectified in the revised manuscript.
Question: And do not FC for fuel cell. And not use, LTFC or HTFC.
Answer: The given comment is valid and accepted. The necessary corrections have been carried out in the revised manuscript.
Question: The “best-known proton conductor is PFSA, which consist of a hydrophobic 42 polytetrafluoroethylene (PTFE)” – this is correct, but one should also refer to NafionTM
Answer: We appreciate and comply with the referee’s comments. As per the reviewer suggestion, the information about NafionTM has been discussed and incorporated in the revised manuscript. Furthermore, NafionÒ replaced with NafionTM.
Question: Typo in Fig 1 for OH-
Answer: Thank you very much for identified the mistake. We strongly apologies for the mistake. It was typo error. The manuscript was thoroughly checked for grammatical error/mistakes/typo errors, and it has been rectified in the revised manuscript.
Question: Typo is missing ‘degree’ page 62
Answer: We sincerely thank the reviewer for the kind notice and valuable suggestion. The authors submitted the manuscript in word format, not in a journal template. After submission, the journal office changed the format before sending it to reviewers. Thus, the formats are changed. However, we have rectified the corrections in the revised manuscript.
Question: Page 62 – Most automotive PEMFC will operate at above 100 degrees C in higher pressure.
Answer: We appreciate and comply with the referee’s comments. As per the reviewer suggestion, the operating temperature of PEMFCs has been modified.
Question: Page 82 typo (missing spaces in many locations)
Answer: Thank you very much for identified the mistake. We strongly apologies for the mistake. It was a typo error. The manuscript was thoroughly checked for grammatical errors/mistakes/typo errors, and it has been rectified in the revised manuscript.
Question: Line 269 has off page para from ‘TAB’ is missing
Answer: Thank you very much for identified the mistake. We strongly apologies for the mistake. It was a typo error. The manuscript was thoroughly checked for grammatical errors/mistakes/typo errors, and it has been rectified in the revised manuscript.
Question: OVERALL – the ‘Jargon’ must be set-up with to many typos and the ‘English’ must be improved.
Answer: We sincerely thank the reviewer for the kind notice and valuable suggestion. The authors submitted the manuscript in word format, not in a journal template. After submission, the journal office changed the format before sending it to reviewers. Thus, the formats are changed. However, we have rectified the corrections in the revised manuscript.
Question: Water in PFSA including Schroeder’s paradox; however, it contains a non-thermodynamic assumption of constant water activity (equal to 1) for water contents beyond equilibrium. This should be discussed for a few lines. The Schroeder’s paradox should discuss the implications of this new membrane material.
Answer: We appreciate and comply with the referee’s comment. As per the reviewer suggestion, the Schroeder’s paradox concept has been discussed in the revised manuscript. “Sorption may be measured by bringing a membrane to equilibrium with a liquid by either immersion of the membrane into the liquid (directly) or by contact with the vapor phase (isopiestically). since the solution, the vapor and the sample are all in equilibrium, it is believed that there is no difference between the two methods. The uptake of water by PFSA from a liquid reservoir and a saturated vapor reservoir differs under the same conditions. This phenomenon is called Schroeder’s paradox, and more recently attempts have been made to explain this phenomenon theoretically”.
Question: In ‘light vehicles’ fuel cells are pressurized (and above 100 degree C). This should be discussed in the background.
Answer: We appreciate and comply with the referee’s comments. As per the reviewer suggestion, the above mentioned topic has been discussed in the revised manuscript.
Question: The chemistry structure should have subscripts.
Answer: We appreciate and comply with the referee’s comments. The necessary corrections have been carried out in the revised manuscript.
Question: For a review paper, this paper has too many typos, references are outdated, and Jargon is weak.
Answer: Thank you very much for identified the mistake. We strongly apologies for the mistake. It was a typo error. The manuscript was thoroughly checked for grammatical errors/mistakes/typo errors, and it has been rectified in the revised manuscript.
Question: Page 351 – I disagree. The Anode use 99.995 hydrogen, so the ‘toxic’ is only a concern for the cathode. This discussion should be improved.
Answer: We appreciate and comply with the referee’s comments. Giving due respect to the reviewer comment, page 351 indicated by you is direct methanol fuel cells (CH3OH and O2 used as the fuel). In DMFC, the carrier ion is the hydrogen ion (H+) and the direction of the flow of the ions is from the anode to the cathode-the same as for PEMFCs. When methanol enters the fuel cell at the anode, however, water is needed to enable the oxidation reaction to occur, which requires precise metering of methanol and water at the anode. One product of this oxidation is carbon dioxide (CO2) which must be vented from the system. The reaction also produces electrons and hydrogen ions (H+). The hydrogen ions pass through the electrolyte while the electrons pass through the external electrical circuit to the cathode. The hydrogen ions, electrons, and oxygen from the air react and produce water (H2O) at the cathode.
Question: Line 512 - “PEMFCs/DMFCs, 511 sulfonated polysulfone and its composites with inorganic fillers, layered double hydroxides, metal organic frameworks have been investigated in this present review.” This is correct, but this has been known for 20 years. The advance of AEM should be updated.
Answer: We appreciate and comply with the referee’s comments. In the revised manuscript, we have provided the advancements of AEM in the conclusion section.
Question: Too many references are very OLD (10 or 20+ old).
Answer: We appreciate and comply with the referee’s comments. Giving due Respect to the reviewer comment, we used a total of 125 references in the present review article. In these references, 62 references are in the year from 2016 to 2021, 28 references from 2010 to 2015, and only 11 references are from before 2000. We would like to bring your kind notice that the review article is a combination of past, present, and future research accomplishments. So, one can’t leave the old references as such. Also, the percentage of old references is significantly less (approximately 15%).
Question: Format for the references out poor presentation.
Answer: The given comment is valid and accepted. The corrections mentioned above have been rectified in the reference section of the revised manuscript.
Question: Even a Review Paper is outdated, and nothing is novel in the analysis. References are outdated.
Answer:
(i) We appreciate and comply with the referee’s comment. Giving due respect to the reviewer’s comment, this present review article elucidates the recent advancements of polysulfone-based membranes and its composites for PEMFC, DMFC, AMFC applications (Year: 2017/2018/2019/2020 and 2021).
(ii) To the best of author’s knowledge, so far, no review articles have been published with the combination of PEMFCs, DMFCs, and AMFCs.
(iii) Furthermore, we discussed the basic differences of cation exchange membrane and anion exchange membranes. Also, in this present review article, we discussed the preliminary characterizations methods such as water uptake, ion exchange capacity, ionic conductivity, methanol permeability, and alkaline stability test in a brief manner.
(iv) Furthermore, in the conclusion section, we clearly mentioned the following paragraph with new insight into the future work for readers and researchers. “As deliberated, the use of AEMs in electrochemical systems could potentially eliminate the common issues such as fuel crossover, confronted in PEMFCs. Additionally, the use of AEMs has several advantages, such as being used in alkaline environments, which enables the use of non-precious metal catalysts. Nevertheless, numerous problems need to be fixed such as poor ionic conductivity (which is accountable for poor voltage efficiency and ohmic losses), insufficient membrane stability in alkaline and oxidative atmospheres, and a lack of suitable alkaline ionomers, especially for AMFCs. Several conventional methods have been extensively studied to improve the ionic conductivity of AEMs. Recently, interpenetrating polymer network (IPN) and pore-enriched composite AEMs have efficiently imitated the Nafion-like morphology, where the hydrophobic polyolefin and the hydrophilic quaternized polymer moiety are well disconnected. As a result, a fabulous enhancement in the ionic conductivity could be attained. Inclusive data regarding the oxidative stability of AEMs can inspire further work towards the modification of existing materials or the development of new materials for AEMs. The development of AEMs based on PEEK, polybenzimidazole, and functional group chemistries based on imidazolium and guanidinium are still in the early stages. Therefore, the chemical stability of these AEMs can be studied in detail, and their performance in electrochemical systems can be explored extensively”.
(v) Further, we strongly believe the current review will be a suitable reference for the researchers, engineers, scientists, and students who would like to pursue their interest in the field of fuel cells using polysulfone-based ion-exchange membranes.
We thank the reviewer for considering our work and for their valuable feedback.

Round 2
Reviewer 1 Report
My previous comments have been carefully considered by the authors and corresponding modifications have been replied one by one. I consider the manuscript is acceptable in its present form.
Reviewer 2 Report
The review comments have been addressed.